# Efficacy and Safety of Native and Recombinant Zona Pellucida Immunocontraceptive Vaccines Formulated with Non-Freund’s Adjuvants in Donkeys

**DOI:** 10.3390/vaccines10121999

**Published:** 2022-11-24

**Authors:** Hilari French, Lorenzo Segabinazzi, Brittany Middlebrooks, Erik Peterson, Martin Schulman, Robyn Roth, Michael Crampton, Anne Conan, Silvia Marchi, Trevor Gilbert, Darryn Knobel, Henk Bertschinger

**Affiliations:** 1Department of Clinical Sciences, Ross University School of Veterinary Medicine, P.O. Box 334, Basseterre 00334, Saint Kitts and Nevis; 2Veterinary Population Management Laboratory, Section of Reproduction, Department of Production Animal Studies, Faculty of Veterinary Science, University of Pretoria, Pretoria 0002, South Africa; 3Council for Scientific and Industrial Research, Pretoria 0184, South Africa; 4Department of Veterinary Tropical Diseases, Faculty of Veterinary Science, University of Pretoria, Onderstepoort 0110, South Africa

**Keywords:** contraception, infertility, asinines, ovarian suppression, fertility

## Abstract

This study aimed to test zona pellucida (ZP) vaccines’ immunocontraceptive efficacy and safety when formulated with non-Freund’s adjuvant (6% Pet Gel A and 500 Μg Poly(I:C)). Twenty-four jennies were randomly assigned to three treatment groups: reZP (*n* = 7) received three doses of recombinant ZP vaccine; pZP (*n* = 9) received two doses of native porcine ZP; and Control group (*n* = 8) received two injections of placebo. Jennies were monitored weekly via transrectal ultrasonography and blood sampling for serum progesterone profiles and anti-pZP antibody titres. In addition, adverse effects were inspected after vaccination. Thirty-five days after the last treatment, jacks were introduced to each group and rotated every 28 days. Vaccination with both pZP and reZP was associated with ovarian shutdown in 44% (4/9) and 71% (4/7) of jennies, 118 ± 33 and 91 ± 20 days after vaccination, respectively (*p* > 0.05). Vaccination delayed the chances of a jenny becoming pregnant (*p* = 0.0005; Control, 78 ± 31 days; pZP, 218 ± 69 days; reZP, 244 ± 104 days). Anti-pZP antibody titres were elevated in all vaccinated jennies compared to Control jennies (*p* < 0.05). In addition, only mild local injection site reactions were observed in the jennies after treatment. In conclusion, ZP vaccines formulated with non-Freund’s adjuvant effectively controlled reproduction in jennies with only minor localised side effects.

## 1. Introduction

Despite a paucity of evidence-based reports, feral or semi-feral donkey populations are perceived to be problematic, particularly in developing countries where they are associated with environmental degradation, resource competition with livestock and wildlife for feed, crop damage and human–animal conflicts [1,2]. Similar negative perceptions have been experienced with African elephants (*Loxodonta africana*) in southern Africa [3], feral horses [2] and white-tailed deer [4] in the USA, and various other so-called nuisance species around the world [5,6].

Other than exclusion or place-aversion methods [5,6,7,8], which are difficult to implement under extensive conditions, the only option to mitigate such conflict is population control. Previously employed population control methods include culling, translocation and fertility control. Although culling was employed to control donkeys in Australia [7] and elephants in Africa [9], this method was largely discontinued because of international pressure, especially from conservation and animal welfare groups. Furthermore, indiscriminate removal of animals has been shown to stimulate density-dependent increases in reproductive rates [10,11].

Fertility control provides the solely effective method to slow-down or fully prevent population expansion. Methods of fertility control include surgical sterilization, intra-uterine devices, hormonal methods and immunocontraception. An ideal contraceptive should be effective, safe in the target animal, remotely deliverable, economical, and, depending on the species, reversible [12]. Reversibility is essential for most wildlife species, but in feral domestic or so-called ‘pest’ species, reversibility may be contraindicated.

Surgical sterilization is effective but irreversible, expensive and impractical to apply in large populations. Intra-uterine devices require capture and restraint of target animals and its efficacy for population control is currently undefined. Hormonal methods require daily feeding or application via depot injections (progestogens) or subcutaneous implants (Suprelorin^®^, Virbac), both of which are logistically difficult for wild or feral animals. Immunocontraception, on the other hand, offers a feasible approach to population control of such animals. Two widely and successfully applied immunocontraceptive methods are anti-GnRH and native porcine zona pellucida (pZP) vaccines. Anti-GnRH vaccines target endogenous GnRH, and are thus effective in both sexes by controlling the release of both LH and FSH, with subsequent suppression of ovarian [13,14] and testicular function [15,16]. As this negatively affects sexual [17,18] and, in some species, territorial behaviour [6], anti-GnRH vaccines may not always be suitable for population control. The pZP vaccine is only effective in females as the immune response targets the zona pellucida proteins of the target animal [19]. This immunocontraception method has been effective as fertility control in >80 mammalian species [20], with reversibility shown in most species [3,20,21,22]. Reported exceptions, with documented irreversible damage to the ovaries, include mice [23], rabbits [24], dogs [25], sheep [26], goats [27], and some primate species [28]. The most-reported examples of successful application of pZP vaccines for fertility control at the population level are in wild horses [29,30,31,32], white-tailed deer [13] and African elephants [3,33]. The potential application for this purpose in donkeys is supported by promising results [21]. In species where the contraceptive effect is reversible, the purported mechanism of action is inhibition of sperm binding to receptor sites and zona-specific antibody penetration of the zona pellucida capsule [19]. As a result, target animals fail to conceive but continue to cycle normally and retain associated reproductive behaviours [34,35]. More recently, an alternate, or possibly additional, mechanism was described in both mares and jennies. The precise mechanism of this novel effect has yet to be clearly defined. However, the resultant suppression of ovarian activity (ovarian shutdown) is accompanied by anoestrus in these species [21,36,37].

The pZP vaccine is usually administered in formulations with Freund’s modified complete (FMCA) and Freund’s incomplete (FIA) adjuvants for primary and booster immunisations, respectively. Injection site reactions associated with vaccine formulations containing Freund’s adjuvants have been well reported in laboratory animals [38], but mitigating this effect by replacing Freund’s adjuvant with sodium phthalate lipopolysaccharide or saline was associated with weaker antibody responses in monkeys [39] and donkeys [21], respectively. Until recently, injection site reactions associated with these formulations in horses were not reported in the literature. This is probably attributable to the limitations associated with intensive clinical monitoring of free-ranging or wild horses following immunisation. Two clinical trials in domestic horses, however, described local muscle swelling, abscess formation, and resultant lameness in some animals [40,41]. In wild African elephants, another difficult species to monitor post immunisation, scars at the darting sites were visible from a helicopter in many cows during follow-up treatments. Lameness, however, has never been reported. The first donkey study reporting the contraceptive effects of pZP formulated with Freund’s adjuvants [42] did not report any injection site reactions. This was, similarly, possibly due to limitations of clinical monitoring during the post-treatment periods. In contrast, severe local injection site reactions, many of which were associated with lameness, were noted in a recent donkey study with jennies immunised with either pZP or recombinant zona pellucida vaccines formulated with Freund’s adjuvants. Although both vaccines were successful in inducing infertility, the injection side effects prohibited further use of Freund’s adjuvants in donkeys.

In a recent study, alternative adjuvants (Addavax, Quil A, Quil A and Poly (I:C), Pet Gel A, and Pet Gel A and Poly (I:C)) for pZP were tested in geldings [40]. From this list of adjuvants or adjuvant combinations tested, the combination of Pet Gel A and Poly (I:C) showed temporal antibody titres that were most similar to those reported for mares vaccinated with pZP formulated with Freund’s adjuvants. In addition, the side effects observed with this formulation were confined to local injection site swelling and mild temperature changes, both of which resolved within three days of treatment. Pet Gel A is a microgel particulate adjuvant that acts as an antigen carrier and depot, and cell and non-cell mediated potentiator [43], whereas Poly (I:C) is a synthetic TLR-3 agonist consistently shown to be amongst the strongest Th1-inducing immunomodulators [44,45]. Th1 and Th2-mediated responses have been suggested as an approach to optimise vaccine immune responses [46,47], which may explain the successful induction of effective anti-ZP antibody titres following the immunization of mares with pZP and reZP vaccines formulated in Pet Gel A and Poly (I:C) [40]. Although pregnancy outcome was not assessed in that study, ovarian shutdown and a decrease in serum anti-Müllerian hormone concentrations were features in treated compared to control mares [36]. Therefore, the goal of the present study was to evaluate the efficacy and safety of pZP and reZP vaccines formulated with the non-Freund’s adjuvants Pet Gel A and Poly(I:C) in Caribbean jennies.

## 2. Materials and Methods

This study was reviewed and approved by Ross University School of Veterinary Medicine (RUSVM) Institutional Animal Care and Use Committee (Protocol #3.17.14). The study was conducted at RUSVM on the island of St. Kitts in the federation of St Kitts and Nevis, West Indies (17°17′34.7604″ N 62°45′31.8024″ W) from July 2018 to May 2019.

### 2.1. Animals and Management

Twenty-four pregnant jennies (mid to late term gestation; determined by transabdominal ultrasound [US]), were sourced from the island of Nevis. On capture, all appeared to be clinically healthy and in good body condition [48], weighing 100 to 140 kg. Estimated ages (according to incisor appearance on dental inspection [49]) varied from three to 13 years. The history of individual animals was unavailable. The jennies were subsequently transported to St. Kitts via water ferry. Upon arrival at RUSVM, they were microchipped and identified with individual numbered collars. They were housed on grass pasture with *ad lib* fresh, clean water and supplemented daily with fresh-cut Guinea grass (*Megathyrsus maximus*) and trace mineral salt blocks. During the following 10 months, they were allowed to foal naturally and acclimatise to their new environment. In this period, they were also halter-trained and habituated to restraint within horse stocks modified for donkeys to allow for safe and effective transrectal examination and US. The foals were separated from the jennies approximately three months before the study commenced.

### 2.2. Study Design

Inclusion criteria included clinical health, non-pregnant status, and a normal reproductive tract with active ovaries. Jennies were examined weekly by transrectal palpation and US (5 mHz linear probe, Sonosite, Universal, Vista, CA, USA) to monitor ovarian activity including follicular and luteal status. Once each jenny had completed at least one oestrous cycle, the animals were allocated randomly to one of three experimental groups as follows: reZP (*n* = 7); pZP (*n* = 9) and Control (*n* = 8) groups. In the reZP group, jennies were treated with three doses of reZP vaccine, whereas two doses of the pZP vaccine or placebo (adjuvant only) were administered in jennies in the pZP and Control groups, respectively. The groups were housed in separate, adjacent outdoor grass paddocks. The reZP group was treated with a primary dose on Day −70 (V1) followed by boosters on Days −35 (V2) and 0 (V3). The pZP and Control groups were treated on Days −35 and 0 with the primary dose (V1) and booster (V2), respectively. Day 0 of the study was defined as the day of last treatment, and on Day 35, one jack (proven fertility) was introduced into each group of jennies until the end of the study. Jacks were rotated between groups every 21 days. Jennies exited the study upon diagnosis of pregnancy. The study was terminated on Day 427 (Figure 1).

### 2.3. Treatments

#### 2.3.1. Preparation of Vaccines

All three vaccine formulations were prepared in the Veterinary Population Management Laboratory, Section of Reproduction, Department of Production Animal Studies, Faculty of Veterinary Science, University of Pretoria, following which they were lyophilised and transferred to RUSVM in St Kitts.

The recombinant porcine ZP3 and ZP4 proteins, supplied by the Council for Scientific and Industrial Research, South Africa, were expressed in *E. coli* according to Gupta et al. [50] with several modifications. The reZP vaccine used in the current study comprised recombinant porcine ZP3 (amino acid (aa) residues 20–421) and recombinant ZP4 (aa residues 23–463), containing the promiscuous T-cell tetanus (aa residues 830–844) and bovine RNase (aa residues 94–104) epitopes at the N-terminus, respectively. The recombinant ZP3 and ZP4 proteins were analysed via SDS-PAGE gels and confirmed by LC-MS peptide mapping. The ZP3 with the TT epitope had 93.3% coverage at 95% confidence, while ZP4 with the bRNase epitope had 92.6% coverage at 95% confidence, analysed with ProteinPilot Software (AB Sciex) [21,36]. The reZP vaccine was prepared in two-dose vials which contained 400 μg of each protein, 6% polymeric adjuvant (Montanide™ PetGel A, Seppic, France) and 1000 μg polyinosinic-polycytidylic acid—TLR3-agonist (Poly(I:C) HMW VacciGrade™, Invivogen, San Diego, CA, USA). The vaccine was the then lyophilized in glass vials and stored at 4 °C. Before administration, each vial was reconstituted with 2 mL sterile water for injection. Each 1 mL dose contained 200 μg of each ZP protein and 500 μg Poly(I:C) in 6%Pet Gel A.

The native pZP was prepared in standard fashion [29]. The pZP vaccine was prepared in two-dose glass vials each containing 200 µg pZP and 1000 µg Poly(I:C) in 6% Pet Gel A, lyophilised and stored at 4 °C. Before administration, each vial was reconstituted with 2 mL sterile water for injection. Each 1 mL dose contained 100 μg pZP and 500 μg Poly(I:C) in 6% Pet Gel A. The adjuvant control was prepared in two-dose glass vials each containing 1000 μg Poly(I:C) in 6% Pet Gel A and sterile saline. After lyophilization they were stored at 4 °C. Before administration each vial was reconstituted with 2 mL sterile water for injection. Each 1 mL dose contained 500 μg Poly(I:C) in 6% Pet Gel A.

#### 2.3.2. Vaccinations

All treatments were administered intramuscularly into the left and right gluteal muscles (1st and 2nd injections, respectively) with an additional injection in the left gluteal for the reZP group (3rd injection in Group reZP). Injections were administered using 3 mL syringes (Monoject™, CardinalHealth™) and 18 gauge 1.5-inch hypodermic needles (Becton Dickinson). The injection sites were clipped and aseptically prepared with 2% chlorohexadine scrub and 70% isopropyl alcohol.

### 2.4. Observations and Sample Collection

#### 2.4.1. Transrectal Monitoring of the Reproductive Tract

Transrectal palpation and US was performed once weekly from Day −70 until Day 427 (end of the study) or the day of pregnancy diagnosis. The left and right uterine horn cross-sectional diameters were measured at their base using internal callipers. Ovarian findings were recorded as the number of visible follicles (approximate resolution threshold ≥ 5 mm) on each ovary and presence of a corpus luteum (CL). Ovarian shutdown was defined as the presence of bilaterally inactive ovaries with only small follicles (<10 mm) and no CL visible on US. Following introduction of jacks, the jennies were monitored for pregnancy.

#### 2.4.2. Monitoring of Injection Sites

Jennies were monitored for five consecutive days following vaccination (Day 0, day of vaccination, through Day 4). Rectal temperature, pulse and respiratory rates were assessed. Injection sites were examined for side effects and scored as follows: Score 0 = no swelling; Score 1 = mild swelling; Score 2 = moderate swelling, and Score 3 = severe swelling. Additionally, signs of lameness or abscess development were recorded. All procedures were performed daily between 06:00 and 08:00 AM.

#### 2.4.3. Collection of Serum Samples

Blood samples were collected by jugular venepuncture in plain evacuated tubes (Terumo™ VENOJECT™ II, Fischer Scientific, Waltham, MA, USA) weekly from Day −70 to the day of pregnancy diagnosis or the end of the study (Day 427). After collection, blood was allowed to clot overnight at room temperature (22 °C) and serum was separated and stored at −80 °C until assayed.

#### 2.4.4. Anti-pZP Antibody Titre Assays

Anti-pZP antibody titres were determined on serum samples collected from all jennies on Days −70, −35, 0, 35, 70, 105, 140, 185, 220, 255, 290, 325, 360, 395, and 427 until either pregnancy diagnosis or study termination on Day 427. Anti-pZP antibody titre assays were performed by enzyme-linked immunosorbent assay (ELISA) at RUSVM, using a modification of a method previously described [22]. Reference serum consisted of pooled samples collected from the pZP group on Day 35, when maximal antibody responses were anticipated. Briefly, 96-well plates (MaxiSorp, cat. no. NUN430341) were incubated at 2–8 °C for 16 h with 1 μg purified pZP in 100 μL coating buffer (2.94% NaHCO_3_, 1.59% Na_2_CO_3_, pH 9.6) per well. Plates were washed four times with phosphate-buffered saline (PBS) containing 0.05% Tween 20 and then blocked with 0.03% bovine serum albumin in PBS for 16 h at 2–8 °C. Plates were then incubated with serial dilutions of reference standards (1:500–1:64,000) and test samples (1:250–1:32,000) in duplicate at 37 °C for 1 h. Wells containing PBS served as blanks. After washing four times, antibodies were detected by incubating plates with 100 μL/well of a 1:10,000 solution (diluted with assay buffer) of a Protein G-horse radish peroxidase conjugate concentrate (1 mg/mL; EMD Millipore Corporation, 28820 Single Oak Drive, Temecula, CA, USA) at 37 °C for 1 h. After washing four times, plates were developed with trimethylene blue (SureBlue™, KPL, Gaithersburg, MD, USA). The reaction was stopped by adding 50 μL of 2 mol/L H_2_SO_4_ per well. Absorbance was measured at 450 nm using a microplate photometer (BioTeck ELx800, Winooski, VT, USA). Sample antibody response (titre) was expressed as the serum dilution rate, which achieved an absorbance 1.

#### 2.4.5. Serum Progesterone Assays

Serum samples were analysed at the Clinical Endocrinology Laboratory of University of California, Davis, CA, USA, using a commercial enzyme immunoassay kit (Arbor Assay, K025-H5; Ann Arbor, MI, USA). Validation was carried out to assess possible matrix effects of donkey samples. Serum pools were prepared from samples collected from jennies during oestrus, dioestrus, and early pregnancy. Each pool was subsequently serially diluted (1:1, 1:2, 1:4, 1:8, 1:16, 1:32) to determine parallelism with the reference standard curve. Four of the six dilutions fell within the linear range of the standard curve, with a CV of 26%, indicative of parallelism [51]. Thereafter, the assay was conducted according to the manufacturer’s protocol, using 50 μL of serum diluted 1:16 (non-pregnancy samples) or 1:32 (pregnancy samples) with assay buffer. Progesterone concentrations that exceeded the high range of the reference standard were further diluted and re-analysed as necessary. Within-assay CV and sensitivity were 17% and 0.05 ng/mL, respectively. Following progesterone analysis, the inter- and intra-assay coefficients of variation were calculated to be 7.03% and 7.93%, respectively.

### 2.5. Data Analysis

Data analyses of ovarian and uterine features, intervals to ovarian shutdown and pregnancy, interval between intervals to ovarian shutdown and pregnancy, progesterone concentrations, titres and adverse events were performed with GraphPad Prism 8.0.1. (GraphPad Software, San Diego, CA, USA). Data that were not normally distributed according to Kolmogorov–Smirnov tests were transformed to natural logarithms. The intervals to ovarian shutdown and pregnancy, the number of ultrasonographically visible follicles, uterine diameter, and progesterone concentrations were evaluated using a mixed model and Tukey’s post hoc test. Titres of jennies presenting with ovarian shutdown and jennies without ovarian shutdown in each group were evaluated by *t*-test. Scores of adverse events were tested by Kruskal–Wallis test followed by Dunn’s test. Significance was set at *p* ≤ 0.05 for all tests. All data were presented as mean ± SD unless otherwise stated.

## 3. Results

### 3.1. Effects of Treatments on Uterine Features, Ovarian Function, and Pregnancy Rates

Compared to Day −70, the jennies vaccinated with pZP or reZP showed a decrease in the mean follicle number (≥5 mm) observed on Day 98 (*p* = 0.007; Figure 2A). Similarly, the mean uterine diameter of the two groups was smaller on Day 196 compared to Day −70 (*p* < 0.05; Figure 2B).

Four of seven reZP and four of nine pZP jennies demonstrated ovarian shutdown characterised by inactive ovaries during the study. The mean interval from Day 0 to shutdown was 118 ± 33 and 91 ± 20 days for reZP and pZP jennies (*p* > 0.05), respectively. The duration of ovarian shutdown was 75 ± 59 and 146 ± 92 days for reZP and pZP jennies (*p* > 0.05, Table 1), respectively. None of the jennies in the control group experienced any ovarian dysfunction. The progesterone concentrations of jennies during the shutdown period were all low (<2 ng/mL) in the pZP group, whereas only two (50%, 2/4) in the reZP group had low progesterone concentrations (<2 ng/mL) during shutdown. The remaining two jennies in the reZP group presenting with ovarian shutdown maintained higher progesterone concentrations (>2 ng/mL) during this period. Progesterone levels were similar in all groups during the study in jennies without ovarian shutdown (*p* > 0.05, Figure 2C). Individual progesterone concentrations are highlighted in Appendix A.

By the end of the study (Day 427), the proportion (95% confidence interval) of jennies pregnant in each group was 100%, 78% and 100% for the reZP (*n* = 7/7), pZP (*n* = 7/9), and Control (*n* = 8/8) groups, respectively. Mean interval to pregnancy from Day 0 was 244 ± 104, 218 ± 69 and 78 ± 31 days for the reZP, pZP and control groups, respectively (*p* = 0.0005, Table 1). Furthermore, of the jennies that entered ovarian shutdown, all four in the reZP and two of four pZP were pregnant by the end of the study period. The interval between the end of ovarian shutdown and pregnancy was 75 ± 84 and 102 ± 89 days for the reZP and pZP jennies, respectively (*p* > 0.05, Table 1). In addition, there were no changes (*p* > 0.05) in the mean interval from Day 0 to pregnancy in jennies of both treatment groups that entered (reZP, 270 ± 114 days; pZP, 305 ± 35 days) or did not enter (reZP, 238 ± 116 days; pZP, 183 ± 28) in ovarian shutdown. Compared to the Control group, pregnancy was delayed by 166 and 140 days in reZP and pZP groups, respectively.

The mean ± SEM titre in each treatment group during the study is highlighted in Figure 3. None of the jennies in the Control group developed anti-pZP antibody titres, whereas the titres in the reZP and pZP groups started to increase on Days −35 and Day 0 (*p* < 0.05), respectively. The anti-pZP antibody titres were higher in reZP and pZP groups than the control group from Day 0 until the end of the study for the Control jennies (Day 105; *p* < 0.05). Peak titres in reZP and pZP treatment groups were reached by Day 35 (Figure 3, Table 2). No difference in mean titres were noted between pZP and reZP groups throughout the study (*p* > 0.05). However, jennies presenting with ovarian shutdown developed higher titres than jennies without ovarian shutdown (Figure 4, Table 2). The likelihood of developing ovarian shutdown increased by 33% for each rise of 1000 in antibody titre (hazard ratio = 1.33, *p* = 0.004; Appendix A). Except for one jenny, all animals exhibited a variable period of shutdown if the titre exceeded a threshold of 3000. In addition, the likelihood of jennies becoming pregnant decreased by 42% for each 1000-increase in Day 35 antibody titre (hazard ratio = 0.58, *p* = 0.0006; Appendix A). The individual timeline to pregnancy and anti-pZP antibody titres is highlighted in Figure 5.

### 3.2. Injection Site Reactions

No changes in pulse, respiratory rates, and rectal temperatures were observed in any the jennies following immunization. Jennies in all groups presented with swelling (2, range 1 to 3) after each treatment. However, no abscesses were diagnosed. Furthermore, no signs of lameness were observed throughout the study.

## 4. Discussion

This study was designed to evaluate the efficacy and safety of zona pellucida vaccines using a non-Freund’s adjuvant formulation in jennies. In a previous study [21], pZP and reZP antigens formulated with Freund’s adjuvants were successfully used to control fertility in jennies. However, an unacceptably high incidence of severe adjuvant-associated injection site side effects, including local swelling, abscessation and lameness were observed in the pZP, reZP and Freund’s adjuvant-only groups [21]. Similar adverse effects have been reported in horses injected with Freund’s adjuvant [52], and injection site reactions associated with vaccine formulations containing Freund’s adjuvants are well documented in laboratory animals [38]. Moreover, a recent study reported an acute stress response, characterised by an increase in faecal corticosteroid metabolite concentrations (FCMC) after the first vaccination with pZP and reZP formulated with Freund’s adjuvants. Following the booster, FCMC remained chronically high [53]. The changes in FCMC were more pronounced in jennies that developed injection site reactions and open abscesses. Corroborating with observations of transient injection site swelling in a previous horse study using ZP vaccines [52], none of the jennies in the present study developed exacerbated side effects (lameness and abscesses) or changes in clinical parameters (pulse, respiratory rates and rectal temperature) following treatment with the vaccines formulated with Pet Gel A and Poly(I:C). This clearly demonstrated the safety of these adjuvants when formulated with pZP and reZP antigens.

In addition, good anti-pZP antibody titres responses have been correlated with reduced fertility in many species [3,13,20,21,23,24,25,28,29,30,31,32,33]. In mares, antibody titres were found to be significant predictors of ovarian shutdown after vaccination with either pZP or reZP [36,40]. Similarly, in a previous study in jennies, a distinct threshold effect of antibody titres on ovarian dynamics and fertility was observed [21]. It is worth noting that the substitution of Freund’s adjuvant with saline in the same study induced a relatively smaller rise in mean anti-pZP antibody titres, but reduced injection site reactions [21]. Satisfactory temporal changes in anti-pZP antibody titres, similar to those found after the use of Freund’s adjuvants, were achieved in mares following the use of the non-Freund’s adjuvants Pet Gel A and Poly(I:C) [40]. In jennies, the present study produced similar results, confirming that Pet Gel A and Poly(I:C) were both safe and effective to use as alternative adjuvants to Freund’s in this species.

The treatment of jennies with non-Freund’s formulated pZP and reZP was associated with reduced ovarian activity in the present study, as observed with Freund’s adjuvanted pZP and reZP [21]. Although previous studies using ZP vaccines in feral horses [29,30,31,54,55] and donkeys [42] failed to observe ovarian shutdown, more recent studies have reported temporary ovarian suppression in mares [22,52] and jennies [21] after treatment with either pZP or reZP. Furthermore, varying levels of ovarian suppression were noted in mares treated with reZP and pZP formulated with the same adjuvant (Pet Gel A and Poly(I:C)) used in the present study [36], showing that the phenomenon is not confined to vaccines formulated with Freund’s adjuvant.

In this study the contraceptive effect was slightly greater in the pZP than in the reZP group, although the differences were not statistically significant. For example, the mean time to shutdown from Day 0 was shorter (91 vs. 118 days) and mean duration of shutdown was longer (146 vs. 75 days) in the pZP than in the reZP treated jennies. Conversely, mean time to pregnancy from Day 0 was longer in the reZP than in the pZP group (244 vs. 218 days). Although reZP used in this study consisted of porcine ZP3 and ZP4 amino acid residues and, compared to native pZP, was devoid of ZP2 and not glycosylated, the anti-pZP antibody concentrations in jennies immunised with reZP were, effectively, not significantly different to the jennies treated with the pZP vaccine. In the previous jenny study, which utilised Freund’s adjuvants, reZP performed marginally better than pZP and mean antibody titres were, once again very similar. To confirm these findings, a similar study using a larger number of jennies is indicated.

The primary contraceptive mechanism of action of ZP vaccines is ascribed to an antibody-mediated interference with sperm-zona binding/penetration and subsequent fertilization. In addition, zona pellucida vaccines have been associated with ovarian dysfunction in various other species [21,37], some instances of which resulted in ovarian damage and permanent infertility [23,24,25,26,27,28]. Temporary ovarian shutdown does not, however, exclude inhibition of sperm zona binding as a ZP-induced mechanism of contraception. Indeed, as reported in mares [22,36] and from a previous jenny study [21], this study showed that animals that continued to cycle also failed to conceive, albeit temporarily. The precise mechanism or mechanisms responsible for the suppression of ovarian function is currently undefined. Previous studies in mares treated with pZP formulated with Freund’s [56] or with reZP formulated with same adjuvants used in this donkey study [36], demonstrated a significant decrease in anti-Müllerian hormone (AMH) concentrations. This decrease was significantly positively correlated with both ovarian size and antral follicle counts. Anti-Müllerian hormone is largely produced by small antral follicles where zona formation is taking place. This could affect granulosa-oocyte communication and possibly factors such as growth differentiation factor-9 and bone morphogenetic protein 15 (BMP 15) [57]. Furthermore, in the present study, ovarian shutdown was associated with higher anti-pZP antibody titres on Day 35. A return to cyclic activity followed by pregnancy, was seen once these titres have waned somewhat. The decrease in antral follicle count was likely due to an immune mediated destruction or malfunction of follicles [52]. However, further investigations are required to more precisely define the mechanism/s of ZP-induced ovarian shutdown.

Another interesting observation in some of the current study’s jennies with ovarian shutdown, was the presence of raised peripheral progesterone concentrations and absence of a visible CL on ultrasound. It is worth noting that progesterone, typically characterised as a gonadal hormone, is also produced by the adrenal glands in response to episodes of stress or pharmacological stimulation of the hypothalamic-pituitary-adrenal (HPA) axis [58,59,60]. Cortisol has been the most useful and reliable parameter to evaluate HPA axis response to stress. Although cortisol was not assessed in the present study, cortisol and progesterone concentrations have been positively correlated with stress or administration of adrenocorticotropic hormone in many species [58,60,61,62,63] including humans [64,65] and ovariectomised rats, dogs and cats [66,67,68]. In donkeys, elevation in cortisol levels have been observed following the presumed stress of severe injection site reactions from inoculation with reZP and pZP formulated with Freund’s adjuvant [53]. While no severe vaccine-associated adverse reactions were observed in the animals of the current study, the raised progesterone concentrations observed in jennies with ovarian shutdown may possibly have been associated with mild vaccine-associated stress episodes. Another hypothesis for the raised progesterone concentrations in jennies with ovarian shutdown is incomplete luteolysis and putative residual luteal tissue undetectable on ultrasonography. If so, this may be explained by the lack of follicular growth and oestradiol levels, although the latter was not investigated in this study [69]. A reduction in total urine oestradiol concentrations was previously reported in pZP-vaccinated mares with ovarian dysfunction [32], and may explain the failure to induce complete luteolysis in jennies in the present study [69]. These hypothesised mechanisms warrant further investigation in donkeys.

The lack of ovarian activity in the present and previous studies [21] may also explain the observed decrease in uterine diameter. The effects of ovarian steroids on uterine function, and thus uterine diameter and endometrial thickness, is well established in mares [70,71]. In the presence of reduced peripheral ovarian steroid concentrations, as found during anoestrus, the uterus appears smaller and irregular [71,72]. Thus, this supports the postulated reduced ovarian activity observed in treated jennies in both this and a previous donkey study [21] that was associated with decreased uterine diameter.

It is apparent from the results that ovarian shutdown accounted, at least partially, for the mechanism of immunocontraception in four jennies of each treated group. The contraceptive mechanism in the remaining five of nine and three of seven pZP and reZP-treated jennies, respectively, was likely due to blocking of sperm zona binding/penetration, thus preventing fertilisation. However, although our results are consistent with previous reports in jennies [21,42] and mares using Freund’s adjuvant [29,30,31,54], it is worth noting that present study was conducted in a controlled population with a limited number of jennies. Therefore, it is possible that different results may be obtained in larger uncontrolled populations. Furthermore, by the end of the study (Day 427), 78% and 100% of the jennies treated with pZP and reZP, respectively, were pregnant, which supports the reversibility of the two immunocontraceptive vaccines in their current formulation. Conversely, it highlighted the need for booster immunisations to maintain the contraceptive effect for management purposes. The booster protocol for pZP has been well established for horses [31,32]. In elephants, two immunisations in Year 1 followed by single annual boosters thereafter, are 100% effective at preventing conception in populations of ≤300 animals [3]. Therefore, further studies are needed to develop a similar protocol for feral donkeys and to address the effect of consecutive years of vaccination with ZP vaccines in this species [31,32].

## 5. Conclusions

In conclusion, pZP and reZP vaccines formulated using the non-Freund’s adjuvants Pet Gel A and Poly(I:C) delayed conception in donkeys and was associated with mild, transient, local injection site side effects. However, the precise mechanism or mechanisms governing the contraceptive effects of these vaccines remain undefined, and further studies are indicated to elucidate the mechanism of ovarian function suppression caused by ZP vaccines.

## Figures and Tables

**Figure 1 vaccines-10-01999-f001:**
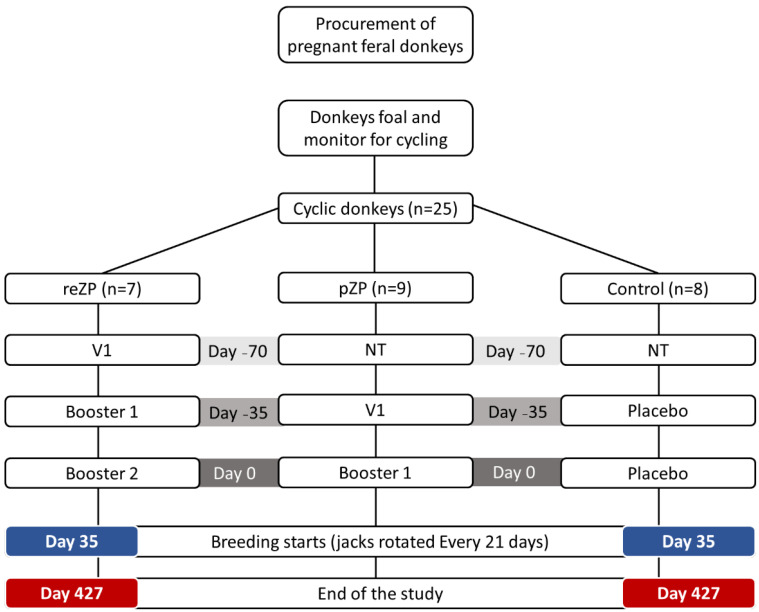
Timeline and treatment protocol for the 24 jennies treated with recombinant zona pellucida vaccine (reZP), porcine zona pellucida vaccine (pZP), or placebo (Control). NT, no treatment; V1, primary vaccination; Booster 1, 1st booster; Booster 2, 2nd booster. Day 0, day of the final vaccination.

**Figure 2 vaccines-10-01999-f002:**
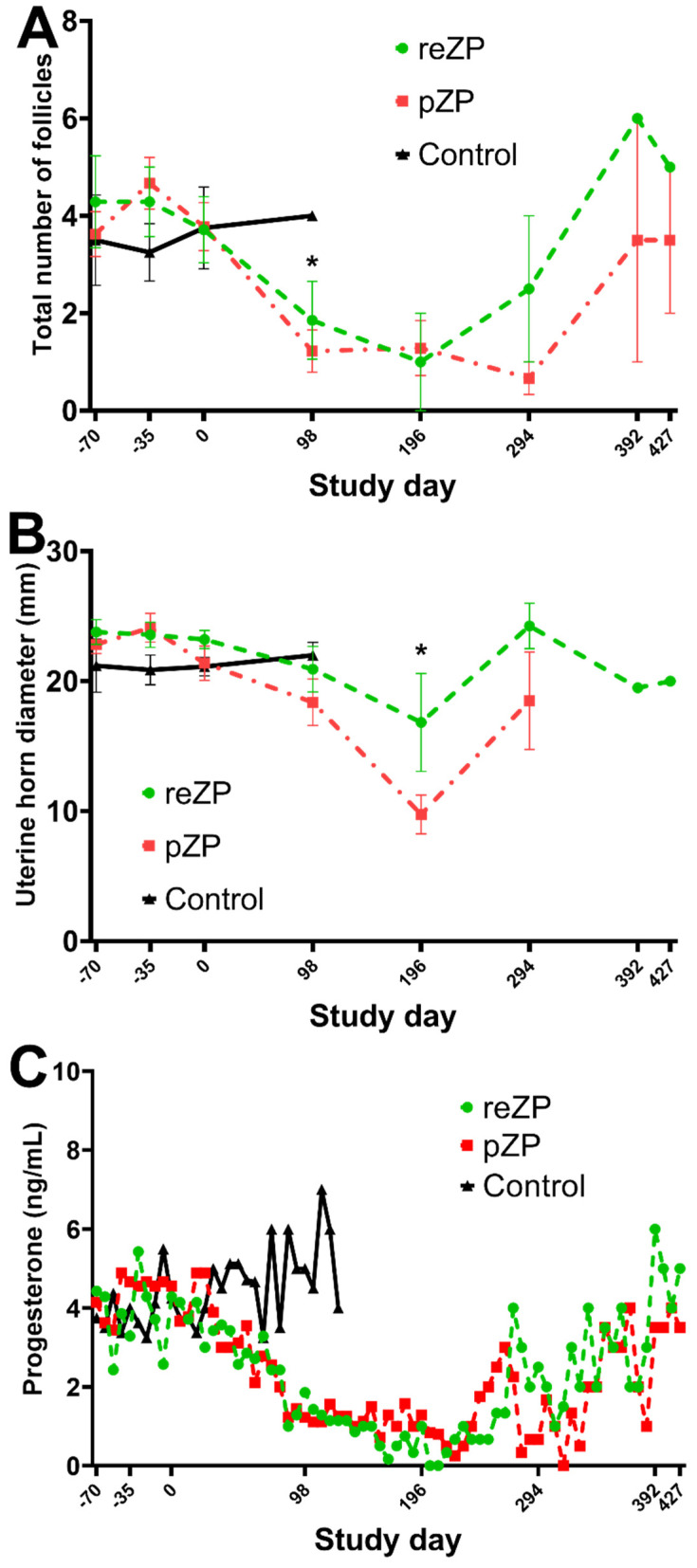
Total mean follicle counts ((**A**); ±SEM), uterine horn diameter ((**B**); ±SEM) and progesterone concentration ((**C**); mean only) of jennies vaccinated with recombinant zona pellucida vaccine (reZP), porcine native zona pellucida (pZP), or placebo (Control) from Day −70 to the day of pregnancy diagnosis. Jennies were vaccinated with three doses of reZP on Days −70, −35 and 0. For both pZP and Control, the primary dose and the booster were administered on Days −35 and 0. V1, primary vaccination; V2, 1st booster vaccination; V3, 2nd booster vaccination). Asterisk (*) denotes effect of time (Day −70; *p* < 0.05).

**Figure 3 vaccines-10-01999-f003:**
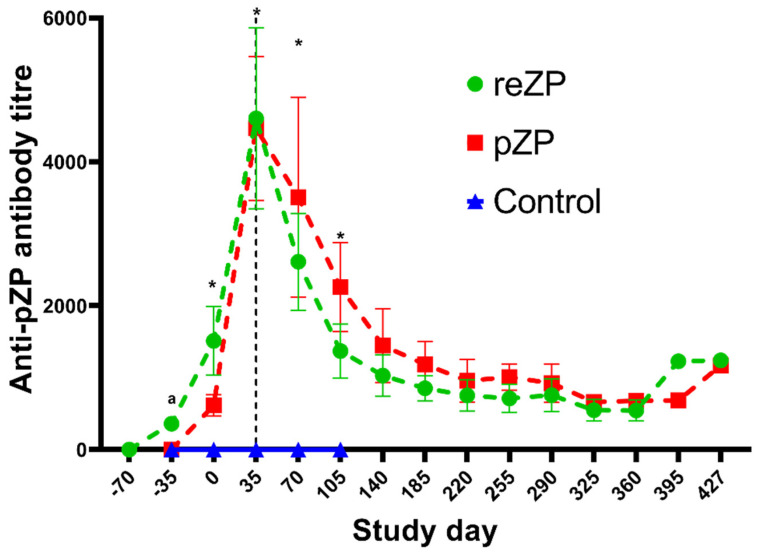
Mean (±SEM) anti-pZP antibody titres (±standard error) of Control, reZP and pZP treated groups post immunisation with Control, reZP and pZP vaccines, respectively. Day 0 = day of last immunisation. Porcine native zona pellucida vaccine (pZP), recombinant zona pellucida vaccine (reZP), or placebo (Control, just the Poly I:C/PetGel A adjuvant). Jennies were vaccinated with three doses of reZP on Days −70, −35 and 0. For both pZP and Control, the primary dose and the booster were administered on Days −35 and 0. Superscript letter (^a^) denote difference between reZP and Control group, wheres asterisk (*) denotes differences between both treated groups (reZP and pZP) and the Control group (*p* < 0.05).

**Figure 4 vaccines-10-01999-f004:**
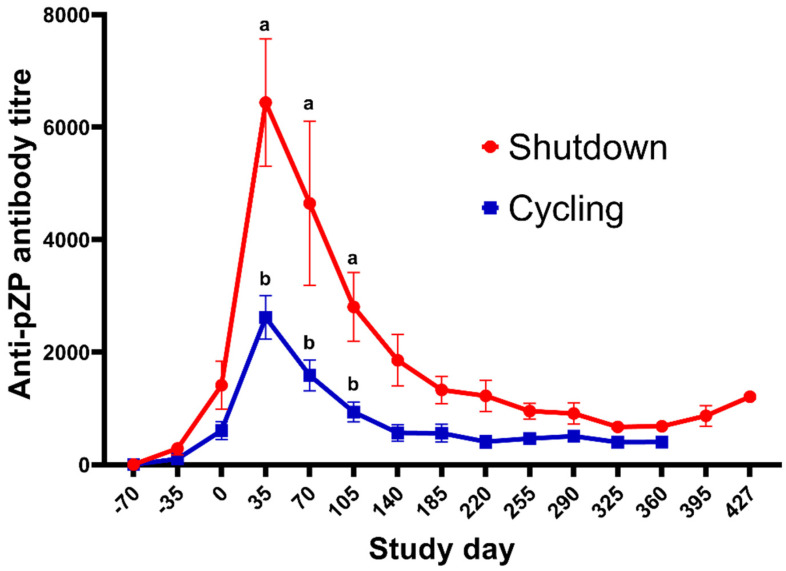
Mean (±SEM) anti-pZP antibody titres (±standard error) of jennies vaccinated with reZP or pZP presenting with (*n* = 8) or without (*n* = 7) ovarian shutdown. Day 0 = day of last immunisation. Jennies were vaccinated with three doses of reZP on Days −70, −35 and 0. For pZP, the primary dose and the booster were administered on Days −35 and 0. Superscript letters (^a,b^) denote significant difference between jennies with or without ovarian shutdown (*p* < 0.05).

**Figure 5 vaccines-10-01999-f005:**
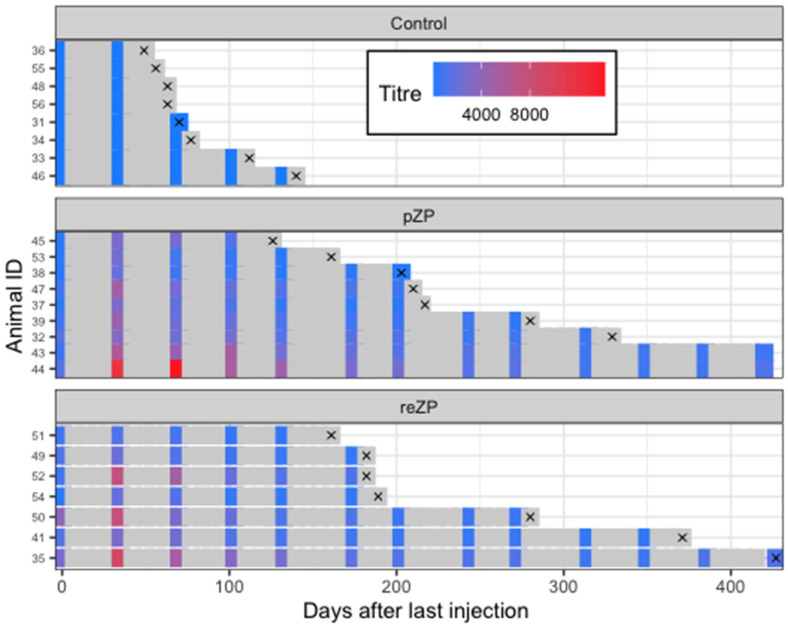
Timelines arranged by treatment groups (Control, reZP and pZP) and individual jennies from day of final immunisation (Day 0) through to study exit. Anti-pZP antibody titres status is indicated by the color (refer to color chart) of the squares on the days of sample collection. The symbol ‘×’ denotes diagnosis of pregnancy.

**Table 1 vaccines-10-01999-t001:** Intervals for ovarian shutdown and pregnancy in jennies vaccinated with recombinant zona pellucida (reZP), porcine native zona pellucida (pZP) vaccines, or placebo (Control).

End Points	reZP (*n* = 7)	pZP (*n* = 9)	Overall Vaccinated Jennies	Control (*n* = 8)
Interval from Day 0 to shutdown	118 ± 33 days (4/7)	91 ± 20 days (4/9)	104 ± 29 days	-
Duration of ovarian shutdown	75 ± 59 days	146 ± 92 days	114 ± 83 days	-
Interval to pregnancy from Day 0	244 ± 104 days (7/7) ^a^	218 ± 69 days (7/9) ^a^	232 ± 87 days (14/16)	78 ± 31 days (8/8) ^b^
Interval between the end of the ovarian shutdown and pregnancy	75 ± 84 days (4/7)	102 ± 89 days (4/9)	88 ± 82 days (8/16)	-

Mean ± SD. Day 0 = day of last immunization. Jennies were vaccinated with three doses of reZP on Days −70, −35 and 0. For both pZP and Control, the primary dose and the booster were administered on Days −35 and 0. Superscript letter (^a,b^) denote difference between groups (*p* < 0.05).

**Table 2 vaccines-10-01999-t002:** Anti-pZP antibodies titres at Day 35 in jennies vaccinated with recombinant zona pellucida (reZP), porcine native zona pellucida (pZP) vaccines, or placebo (Control).

Anti-pZP Antibody	reZP (*n* = 7)	pZP (*n* = 9)	Overall Vaccinated Jennies	Control (*n* = 8)
Titres at Day 35 of the study	4604 ± 3325 ^a^	4465 ± 3001 ^a^	4526 ± 3038 ^a^	0 ^b^
Titres at Day 35 of jennies with ovarian shutdown	6550 ± 3144 (4/7)	6322 ± 3742 (4/9)	6436 ± 3202 ^Y^	-
Titres at Day 35 of jennies with no ovarian shutdown	2008 ± 806 (3/7)	2980 ± 511 (5/9)	2612 ± 1088 ^X^	-

Mean ± SD. Day 0 = day of last immunisation; Day 35 = peak of anti-pZP antibody titres. Jennies were vaccinated with three doses of reZP on Days −70, −35 and 0. For both pZP and Control, the primary dose and the booster were administered on Days −35 and 0. Superscript letter (^a,b^) denote difference between groups, and (^X,Y^) within columns (*p* < 0.05).

## Data Availability

The original contributions presented in this study are included in this article, further inquiries can be directed to the corresponding authors.

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
