# Peer review of "Efficacy and Safety of Native and Recombinant Zona Pellucida Immunocontraceptive Vaccines Formulated with Non-Freund’s Adjuvants in Donkeys"

_vaccines, 2022, doi:10.3390/vaccines10121999_

Round 1

Reviewer 1 Report

This is a well, clear written paper describing study to evaluate the efficacy and safety of zona pellucida vaccines using a non-Freund’s adjuvant formulation in jennies.

Fertility control provides the solely effective method to slow-down or fully prevent 48 population expansion. The goal of the present study was to evaluate the efficacy and safety of pZP and reZP vaccines formulated with the non-Freund’s adjuvants Pet Gel A and Poly(I:C) in Caribbean jennies.

Based on study results authors concluded that pZP and reZP vaccines formulated using the non-Freund’s adjuvants 453 Pet Gel A and Poly(I:C) delayed conception in donkeys and was associated with mild, transient, local injection site side effects.

Author Response

Response to the editor and reviewers:

Dear Editor and peer reviewers,

We are grateful for the opportunity to review our manuscript for consideration in the Vaccines journal. The comments and suggestions made by peer reviewers certainly improved the quality of the manuscript. We have addressed all considerations or provided a rebuttal on few occasions. If there is any remaining concern, we are happy to revisit any point that editor and reviewers considered necessary.

Best regards,

The authors

Reviewer 1

This is a well, clear written paper describing study to evaluate the efficacy and safety of zona pellucida vaccines using a non-Freund’s adjuvant formulation in jennies.

Fertility control provides the solely effective method to slow-down or fully prevent 48 population expansion. The goal of the present study was to evaluate the efficacy and safety of pZP and reZP vaccines formulated with the non-Freund’s adjuvants Pet Gel A and Poly(I:C) in Caribbean jennies.

Based on study results authors concluded that pZP and reZP vaccines formulated using the non-Freund’s adjuvants 453 Pet Gel A and Poly(I:C) delayed conception in donkeys and was associated with mild, transient, local injection site side effects.

Reply: The authors appreciated the Reviewers comments.

Reviewer 2 Report

In this manuscript, the authors show the reversible effects of vaccination without Freund's adjuvant (this is new compared to previous work) against purified or recombinant proteins from the zona pellucida of pigs on the alteration of fertility of female donkeys with or without reduction in the number of ovarian follicles visible on ultrasound (large follicles with antrum). The manuscript is clear, but I have several major and minor questions/concerns.

- What can be the mechanisms explaining the ovarian shutdown in some donkeys? The authors should discuss a very probable disruption of the oocyte/somatic cell dialogue of the ovarian follicles, involving BMP15 and GDF9 in particular.

- The authors must also discuss the fact that the pZP contains ALL the 4 ZP of the zona pellucida (of pig?) as well as the sugars which compose it, not the recombinant zp. Hence the slightly better efficiency of pZP? In addition, there is a difference in structure/sequence between the ZP proteins of different animal species, hence a possible better efficiency of vaccination of donkeys with ZP of donkey?

- What species are the ZP3 and 4 recombinants? Porcine I guess?

- Figure 2: the sem are missing!!!

Author Response

Response to the editor and reviewers:

Dear Editor and peer reviewers,

We are grateful for the opportunity to review our manuscript for consideration in the Vaccines journal. The comments and suggestions made by peer reviewers certainly improved the quality of the manuscript. We have addressed all considerations or provided a rebuttal on few occasions. If there is any remaining concern, we are happy to revisit any point that editor and reviewers considered necessary.

Best regards,

The authors

Reviewer 2

In this manuscript, the authors show the reversible effects of vaccination without Freund's adjuvant (this is new compared to previous work) against purified or recombinant proteins from the zona pellucida of pigs on the alteration of fertility of female donkeys with or without reduction in the number of ovarian follicles visible on ultrasound (large follicles with antrum). The manuscript is clear, but I have several major and minor questions/concerns.

- What can be the mechanisms explaining the ovarian shutdown in some donkeys? The authors should discuss a very probable disruption of the oocyte/somatic cell dialogue of the ovarian follicles, involving BMP15 and GDF9 in particular.

Reply: The precise mechanism that cause suppression of ovarian function is currently undefined. However, the authors appreciated the reviewer perspective and included some hypothesis regarding the involvement of BMP15 and GDF in ovarian function (L418-440).

- The authors must also discuss the fact that the pZP contains ALL the 4 ZP of the zona pellucida (of pig?) as well as the sugars which compose it, not the recombinant zp. Hence the slightly better efficiency of pZP? In addition, there is a difference in structure/sequence between the ZP proteins of different animal species, hence a possible better efficiency of vaccination of donkeys with ZP of donkey?

Reply: It was added in L406-417.

- What species are the ZP3 and 4 recombinants? Porcine I guess?

Reply: It was added in the methodology (L168).

- Figure 2: the sem are missing!!!

Reply: The authors decided to show up only the mean of P4 because the figure would be unclear if the SEM was added. However, the SEM was added for the number of follicles and uterine diameter. To clarify this concern, the individual P4 values are available in the Figure S1, as previously described.

Reviewer 3 Report

In this study French et al. strived to assess the efficiency of an immunocontraceptive vaccine formulated with a non-Freund’s adjuvant in donkeys. The study is scientifically sound and of interest. The research design is appropriate, the methods are well described and the collected data are properly discussed. I have no major comments, however i would recommend the authors to elaborate on possible limitations of their research as well as future prospects for the collected data.

This is truly a well-written manuscript, and there are really very few queries I can come up with. Nevertheless, here are some that may be of value for the study:

- The authors could add some photos of the animals, housing and administration of the vaccine  

- A higher resolution of figure 5 could add more quality to the visual aspect of the data presented in the figure  

- Since a pretty hight age variability amongst the animals was found, the authors could discuss any impact the age has on the efficiency of the vaccine  

- Did the authors study other female hormones besides progesterone? This could add more information to the effect of the vaccine on the ovarian cycle in the animals

Author Response

Response to the editor and reviewers:

Dear Editor and peer reviewers,

We are grateful for the opportunity to review our manuscript for consideration in the Vaccines journal. The comments and suggestions made by peer reviewers certainly improved the quality of the manuscript. We have addressed all considerations or provided a rebuttal on few occasions. If there is any remaining concern, we are happy to revisit any point that editor and reviewers considered necessary.

Best regards,

The authors

Reviewer 3

In this study French et al. strived to assess the efficiency of an immunocontraceptive vaccine formulated with a non-Freund’s adjuvant in donkeys. The study is scientifically sound and of interest. The research design is appropriate, the methods are well described and the collected data are properly discussed. I have no major comments, however i would recommend the authors to elaborate on possible limitations of their research as well as future prospects for the collected data.

Reply: The authors appreciated the reviewer comments and the limitations and future prospects for the results of our study were added (L476-490) as requested.

Round 2

Reviewer 2 Report

Accept.